# Does Caffeine Supplementation Associated with Paralympic Powerlifting Training Interfere with Hemodynamic Indicators?

**DOI:** 10.3390/biology11121843

**Published:** 2022-12-16

**Authors:** Jainara Lima Menezes, Felipe J. Aidar, Georgian Badicu, Stefania Cataldi, Roberto Carvutto, Ana Filipa Silva, Filipe Manuel Clemente, Claudia Cerulli, Joseane Barbosa de Jesus, Lucio Marques Vieira-Souza, Eliana Tranchita, Ciro José Brito, Francesco Fischetti, Gianpiero Greco

**Affiliations:** 1Graduate Program of Physiological Science, Federal University of Sergipe (UFS), São Cristovão 49100-000, Brazil; 2Group of Studies and Research of Performance, Sport, Health and Paralympic Sports (GEPEPS), Federal University of Sergipe (UFS), São Cristovão 49100-000, Brazil; 3Graduate Program of Physical Education, Federal University of Sergipe (UFS), São Cristovão 49100-000, Brazil; 4Department of Physical Education and Special Motricity, Faculty of Physical Education and Mountain Sports, Transilvania University of Braşov, 500068 Braşov, Romania; 5Department of Translational Biomedicine and Neuroscience (DiBraiN), University of Study of Bari, 70124 Bari, Italy; 6Escola Superior Desporto e Lazer, Instituto Politécnico de Viana do Castelo, Rua Escola Industrial e Comercial de Nun’Álvares, 4900-347 Viana do Castelo, Portugal; 7Research Center in Sports Performance, Recreation, Innovation and Technology (SPRINT), 4960-320 Melgaço, Portugal; 8Research Centre in Sport Sciences, Health Sciences and Human Development, Quinta de Prados, Edifício Ciências de Desporto, 5001-801 Vila Real, Portugal; 9Instituto de Telecomunicações, Delegação da Covilhã, 1049-001 Lisboa, Portugal; 10Laboratory of Physical Exercise and Sport Science, Department of Exercise, Human and Health Sciences, University of Rome Foro Italico, Piazza Lauro de Bosis 15, 00135 Rome, Italy; 11Physical Education Course, State University of Minas Gerais, Passos 37900-106, Brazil; 12Department of Physical Education, Federal University of Juiz de Fora, Governador Valadares 35010-180, Brazil

**Keywords:** paralympic powerlifting, resistance training, nutritional supplementation, caffeine, hemodynamics

## Abstract

**Simple Summary:**

Strength training causes benefits related to blood pressure and heart rate. However, the consumption of caffeine, used to improve performance in high-performance sports, can interfere with its effects and benefits. The study researches the interference of caffeine consumption during bench press exercise. The results of the study showed that, despite raising blood pressure, it tended to fall after 24 h of exercise, demonstrating the safe use of this supplement in adapted strength sports.

**Abstract:**

Exercise, including resistance exercise with high loads, has positive hemodynamic responses such as reduced systolic blood pressure (SBP), diastolic blood pressure (DBP), heart rate (HR), Pressure Product Rate (PPR), and estimated myocardial oxygen volume (MVO_2_). Caffeine (CA), used to improve performance, tends to interfere with BP and HR. This study aimed to analyze the effects of CA supplementation on hemodynamic indicators in Paralympic weightlifting (PP). The exercise was performed on 14 male athletes (32.4 ± 8.5 years; 81.7 ± 21.9 kg) for three weeks. Two conditions were evaluated: supplementation with CA Anhydrous 9 mg/kg and with placebo (PL). The adapted bench press was used, with 5 × 5 at 80% 1RM. We evaluated BP, HR, PPR, and MVO_2_, before, after, 5, 10, 20, 30, 40, 50, 60 min, and 24 h later. The CA presented higher absolute values in the pressure indicators than the PL, and after 24 h there was an inversion. The HR was higher in the CA and showed a reduction after 10 min. The PPR and MVO_2_ in the CA presented absolute values greater than the PL, and 24 h later there was an inversion. There was no hypotensive effect, but the use of CA did not present risks related to PPR and MVO_2_, demonstrating the safe use of this supplement in adapted strength sports.

## 1. Introduction

During exercise, systolic blood pressure (SBP), diastolic blood pressure (DBP), and heart rate (HR) tend to increase [1], and changes are derived from the central nervous system that stimulate the excretion of catecholamines [2]. After its completion, hemodynamic indicators decrease [3,4,5]. A single bout of resistance exercise can lower blood pressure [6]. SBP can reduce from 2 to 17 mmHg and DBP from 2 to 7 mmHg [7,8,9], such as Post-Exercise Hypotension (PEH) [5,6]. PEH is related to pressure levels at rest, after physical activity, when there would be a reduction in BP in relation to pre-exercise BP [10]. PEH lasts up to 24 h [11,12]. In addition, physical exercise has demonstrated other positive effects on hemodynamic responses, such as a reduction in HR, pressure product rate (PPR), and estimated myocardial oxygen volume (MVO_2_) in conventional and Paralympic athletes [3,4,5], contributing as a form of non-drug treatment for the control of systemic arterial hypertension [3,4,13]. However, exercises that are performed only on the upper limbs may have a lower hypotensive effect, when compared to exercises developed for the lower limbs [14,15]. Paralympic powerlifting (PP) is a modality adapted from conventional powerlifting (CP), a type of resistance exercise, weightlifting modality, in which the athlete performs squat, bench press and deadlift exercises, working with loads close to one repetition maximum (1RM) [3,5]. In PP, men and women with lower limb disabilities compete (amputees, spinal cord injuries, among others) [16], using only the bench press exercise, in which the athlete has three attempts to perform the movement, with the highest weight raised recorded as the final result [16].

It is common for athletes, both conventional and Paralympic, to consume supplements to improve performance [7,17,18]. Among the supplements, caffeine (CA) is an ergogenic agent widely used to improve the performance of high-intensity physical exercises [19,20]. Moreover, following the removal of CA from the World Anti-Doping Agency’s list of prohibited supplements, consumption of CA in competitions has increased significantly from 2004 to 2015 [19]. However, it can interfere with BP and HR [21,22,23], raising concerns about its use, since people with lower limb deficiencies tend to have more risk factors for arterial hypertension [15]. This is especially relevant for Paralympic athletes, as 12% of them have cardiovascular abnormalities and 2% have a high risk of sudden cardiac death in addition to other heart diseases [4]. CA is an antagonist of adenosine A1 and A2 receptors, capable of crossing the blood–brain barrier and, at the level of the central nervous system (CNS), it stimulates the release of noradrenaline, dopamine, acetylcholine, and serotonin, in addition to stimulating muscle contraction and lipolysis [21,22,23]. Regarding the dose, low doses of CA ≤ 3 mg/kg of body weight have been suggested to be more effective in long-term exercise, while higher doses of 5–9 mg/kg of body weight may be more effective for a healing effect. Performance increase in exercises with high loads [19], as in PP, however, has not yet been investigated in this population.

Studies are inconsistent regarding the exercise-related hemodynamic effects of CA [24,25]. Previous research has investigated the impact of CA supplementation on sports hemodynamics, showing that CA and supplements with this substance tend to produce negative post-exercise hemodynamic responses compared to placeboes, such as increased BP and HR [26,27]. On the other hand, according to a study by Materko et al. [17], during muscle strength exercise, there were no significant increases in HR or BP. In the same direction, in another study, the consumption of CA did not change cardiovascular behavior, both at rest and after a resistance exercise session [24]. In resistance exercises of the CP type, especially in PP, no studies were found to clarify the impacts of the use of CA on hemodynamics. In view of the increase in the consumption of this supplement in PP and the lack of scientific evidence that demonstrates safety in the use of the supplement by these athletes, the present study analyzed the effects of caffeine supplementation on hemodynamic indicators in Paralympic powerlifting. For this, it was hypothesized that caffeine would increase blood pressure or caffeine would not increase blood pressure or caffeine would decrease blood pressure or even acute caffeine consumption in Paralympic powerlifting would or would not cause a risk of vascular overload.

## 2. Materials and Methods

### 2.1. Study Design

This study was a randomized, crossover, blinded, counterbalanced design, where each participant participated as their own control. Participants were randomized to two experimental conditions: supplementation of caffeine anhydrous capsules and a placebo product. By drawing lots, 50% of them were supplemented with caffeine (9 mg/kg) and 50% with a placebo. The study lasted three weeks with an interval of seven days for each session (Figure 1). In the first week, the subjects underwent body assessment, exposure to the 1RM test, signing of the Free and Informed Consent Term (FICT) and rest. In the second and third weeks, supplementation with CA or PL and exercise sessions took place. In both sessions, five sets of five repetitions (5 × 5) were performed with a load of 80% of 1RM and 1 minute of rest between sets for the bench press exercise, as previously described by Austin and Mann [28]. At rest, they returned to their daily activities and were prohibited from performing physical exercises. There was no need for a washout week, as the elimination half-life of the CA varies between 2.5 and 10 h; we believed that this period of seven days was sufficient to remove the CA from the athletes’ systems [29].

### 2.2. Participants

Fourteen Paralympic Weightlifting male athletes, participants in the extension project of the Federal University of Sergipe, Brazil, with at least 18 months of exercise and who were among the top 10 within the ranking of their respective categories at the national level, participated in this study. All fulfilled the necessary prerequisites of the Brazilian Paralympic Committee and are able to compete in the modality [16]. Among the deficiencies: one had atrophy of the left leg, one had an amputation below the right knee, two had an amputation below the right leg knee, one had a traumatic brain injury, one had a lesion in the right leg, one had a spinal cord injury due to schistosomiasis infection, one had an amputation above the right knee, three had sequelae due to poliomyelitis in the lower limbs, and three had arthrogryposis.

As exclusion criteria, the fact that they were using some type of illicit ergogenic resource and had some type of symptomatic cardiorespiratory or cardiac disease was considered. Female athletes did not participate, because in women, alterations in CA metabolism can occur, which may lead to a reduction in the ergogenic effects of CA, due to the interaction of the supplement with female sex hormones [19], especially in women who use contraceptives [23]. The athletes participated in the study voluntarily and signed an informed consent form, under the resolution 466/2012 of the National Research Ethics Commission—CONEP, of the National Health Council, following the ethical principles expressed in the Declaration of Helsinki (1964, reformulated in 1975, 1983, 1989, 1996, 2000, 2008, and 2013) from the World Medical Association. This study was approved by the Research Ethics Committee of the Federal University of Sergipe, CAAE: 2,637,882 (approval date: 7 May 2018). The sample is characterized in Table 1.

### 2.3. Instruments

Body mass was measured seated on an electronic wheeled scale (Micheletti, São Paulo, Brazil), of the digital electronic platform type, with a maximum weight capacity of 3000 kg (dimensions 1.50 m by 1.50 m). To perform the bench press exercise, an official 2.10 m long straight bench (Eleiko, Halmstad, Sweden) and an Olympic barbell (220 cm total length, weight 20 kg), both approved by the International Paralympic Committee [16], were used. BP and HR were assessed using an automatic non-invasive blood pressure monitor (3AC1-1PC, Microlife, Widnau, 146 Switzerland) [4,30].

#### 2.3.1. Caffeine Anhydrous (CA) and Placebo (PL)

Capsules of CA anhydrous and a product PL were made in a compounding pharmacy. The capsules were manufactured in the same size in blue color, those of CA were manufactured according to the body weight of each subject, in amounts of 9 mg per kg of body weight [31,32], while the PL consisted of a capsule containing the necessary dosage of pharmaceutical talc to have an appearance identical to the caffeine capsule. The administration took place by drawing lots immediately after the athletes arrived at the collection site. The time between ingestion of the supplement and the beginning of the tests was 60 min.

#### 2.3.2. Load Determination

The 1RM test was performed to determine the exercise load, using the protocol proposed by Fleck and Kraemer [33]. We asked each participant what weight they expected to lift just once using their maximum effort. Subsequently, more weight was added, until it reached the maximum load that the athlete could lift at once. If the practitioner could not perform a single repetition, 2.4 to 2.5% of the load used in the test was subtracted. Subjects rested between 3 and 5 min between trials. The test did not exceed five tries.

#### 2.3.3. Blood Pressure and Heart Rate Measurement

SBP, DBP, and HR were measured before, immediately after, and at 5, 10, 20, 30, 40, 50, 60 min, and 24 h after each exercise session. MBP was assessed using the equation (DBP = DBP + [SBP − DBP]/3). The pressure product rate was evaluated according to the following equation: PPR = HR × SBP [3,4]. To estimate myocardial oxygen volume, a mathematical function based on the high correlation between PPR and estimated MVO_2_ was used. The following equation expressed in mlO_2_/100 g ventilation/min (VE/min) was used, estimated MVO2 = (PPR × 0.0014) − 6.37 [3,4,34]. All BP measurements were taken on the left arm and the cuff was fixed to the arm with approximately 2.5 cm of distance between its lower extremity and the antecubital fossa [3,4,35]. For this, the subjects remained at rest for 10 min, sitting, and in a calm environment.

#### 2.3.4. Side Effects

All side effects were measured using a side effects questionnaire (QUEST), which is a nine-item measure with a dichotomous (yes/no) response scale of caffeine intake (headache, abdominal/gastrointestinal pain, muscle pain, increased energy and strength, tachycardia/palpitations, insomnia, improved performance, increased urine production and elimination, increased worry/anxiety), and was based on previous publications on side effects derived from caffeine ingestion [31,32,36]. It was applied immediately after the end of the tests and after 24 h.

### 2.4. Procedures

All test sessions were performed in the morning between 9:00 a.m. and 1:00 p.m. for all participants. Before the tests, the athletes were instructed to abstain from the consumption of alcohol and tobacco. In addition, they were instructed not to use any medication or food supplement, in addition to other ergogenic substances during the three weeks of the tests, including coffee and any other product that contains caffeine; tea, cola, chocolate, and energy drinks. This was confirmed through an interview prior to the interventions. When participants arrived at the collection site, they were administered CA or PL according to randomization, subsequently. Soon after, they were asked to remain in reserve and seated for 10 min, and then their BP and HR were measured. Once this was completed, the subjects were asked to stay at rest for another 50 min. After this period, the warm-up started: (shoulder abduction with dumbbells, shoulder press on the machine, rotation of the shoulders with dumbbells) with a series of 20 repetitions in approximately 10 min, followed by a warm-up on the bench press using only the bar (20 kg) without extra weight, where 10 slow reps (3.0 × 1.0 s, eccentric × concentric) and 10 fast repetitions (1.0 × 1.0 s, eccentric × concentric) were performed. Then, an exercise session was performed, characterized by isoinertial loads of 80% of 1RM for concentric and eccentric movements. Athletes were trained to perform the usual speed of the exercise they normally practice and to maintain an equal learning time for the two methods. At the end of the 5 × 5 exercise, BP and HR were measured immediately after, 5, 10, 20, 30, 40, 50, 60 min, and 24 h after, as well as the QUEST side effects questionnaire, applied immediately after and 24 h later. In the third week the switch took place, those who used caffeine received a placebo and those who used a placebo received caffeine.

### 2.5. Statistical Analysis

For data analysis, descriptive statistics were applied using measures of central tendency, mean (X) ± Standard Deviation (SD), 95% confidence interval (95% CI), and coefficient of variation (CV). Statistical analysis was performed using the computerized Statistical Package for the Social Science (SPSS), version 22.0. To verify the normality of the variables, the Shapiro–Wilk test was used, considering the sample size. For evaluation through the questionnaire, the frequencies were used. To evaluate the performance between the groups, the ANOVA (Two-Way, 2 × 2, Moment and Condition) repeated measures and Bonferroni Post-hoc tests were performed. The significance level adopted was *p* < 0.05. The effect size was determined by the values of “partial square eta” (η^2^p), considering values of low effect (≤0.05), medium effect (>0.05 to 0.25), large effect (>0.25 to 0.50), and very large effect (>0.50) [35]. To verify the effect sizes of the paired comparison, Cohen’s "d" was used. A “d” value < 0.2 was considered a trivial effect, 0.2 to 0.6 a small effect, 0.6 to 1.2 a moderate effect, 1.2 to 2.0 a large effect, 2.0 to 4.0 a very large effect, and ≥4.0 an extremely large effect. Cohen’s “d” was calculated as the difference between the mean divided by the pooled SD to estimate the effect size for between-lift comparison [37].

## 3. Results

It was found that immediately and 24 h after the tests using CA, the athletes had low side effects, increased energy and strength, and improved performance (Table 2).

The results of SBP, DBP, MBP, HR, PPR, and estimated MVO_2_ are shown in Figure 2, Figure 3 and Figure 4.

Regarding systolic blood pressure (Figure 2A) it was found that there was a greater response in 24 h between exercise with PL (133 ± 13, 95% CI 125.12–140.02, CV 10%) and with CA (125 ± 12, 95% CI 118.43–132-43, CV 10%, “*” *p* = 0.048, in Condition, η^2^p = 0.171, Medium Effect; “Cohen’s d” = 2.73 Very Large Effect). In the other moments, with the exception of 24 h, there were no differences; however, in relation to the absolute values, the use of caffeine maintained values higher than the placebo.

In (Figure 2B) in relation to diastolic blood pressure, there was a greater response in the exercise with CA between 24 h (77 ± 11, 95% CI 70.25–83.03, CV 14%) and the moments 20 min later (87 ± 12, 95% CI 80.27–94.45, CV 14%, “a” *p* = 0.037; “Cohen’s d” = 3.90 Very Large Effect), 40 min later (85 ± 9, 95% CI 80.18–90.25, CV 10%, “b” *p* = 0.037; “Cohen’s d” = 1.70 Large Effect) and between the moment 50 min later (85 ± 13, 95% CI 77.76–92.38, CV 15%, “c” *p* = 0.023, in moment, η^2^p = 0.178, Medium Effect; “Cohen’s d” = 3.19 Very Large Effect). In the other moments, there were no differences between the use of caffeine and thr placebo.

Mean blood pressure (Figure 3A) showed greater response 20 min later to PL (100 ± 14, 95% CI 91.32–107.97 CV 14%) and CA exercise (104 ± 12, 95% CI 97.04–111.44, CV 12%, in Condition, “*” *p* = 0.036, η^2^p = 0.147, Medium Effect; “Cohen’s d” = 1.39 Large Effect). In the other moments, there were no differences between the use of caffeine and the placebo.

There was a greater response in heart rate (Figure 3B) in exercise with CA between the moments before (76 ± 13, 95% CI 68.91–85.80 CV 17%) and 5 min later (90 ± 14, 95% CI 82.44–98.13, CV 15%, “a” *p* = 0.023; “Cohen’s d” = 5.14 Extremely Large Effect), and 40 min (89 ± 13, 95% CI 81.12–96.03, CV 15%, “b” *p* = 0.031, “a and b” in Moment, η^2^p = 0.168, Medium Effect; “Cohen’s d” = 4.54 Extremely Large Effect). There was also a distinguished response among PL and CA at 10 min (PL 84 ± 9, 95% CI 78.27–88.88 CV 11%, and CA 90 ± 15, 95% CI 81.17–98.55, CV 17%, “*” *p* = 0.033; “Cohen’s d” = 0.99 Moderate Effect), 40 min (PL 82 ± 12, 95% CI 75.16–88.98, CV 15%, and CA 89 ± 13, 95% CI 81.12–96.03, CV 15%, “**” *p* = 0.018; “Cohen’s d” = 2.45 Very Large Effect) and 60 min (PA 81 ± 12, 95% CI 75.55–86.30, CV 12%, and CA 86 ± 13, 95% CI 78.43–93.68, CV 15%, “***” *p* = 0.038; in Interation, η^2^p = 0.341, Large Effect; “Cohen’s d” = 1.79 Large Effect). In the other moments, after the intervention, the absolute values of caffeine were higher than the those of the placebo.

In the pressure product rate (Figure 4A) there was a greater response in exercise with CA between the moments before (10,104 ± 590, 95% CI 8828.06–11,379.37 CV 22%) and after (12,446 ± 686, 95% CI 10,964.94–13,927.37, CV 21%, “a” *p* = 0.001; “Cohen’s d” = 14.75, Extremely Large Effect), 5 min (12,209 ± 657, 95% CI 10,788.64–13629.51, CV 20%, “b” *p* = 0.003; “Cohen’s d” = 14.88 Extremely Large Effect), 10 min (12,200 ± 765, 95% CI 10,547.25–13,852.61, CV 23%, “c” *p* = 0.009; “Cohen’s d” = 9.52 Extremely Large Effect), 20 min (12,367 ± 720, 95% CI 10,811.90–13,922.24, CV 22%, “d” *p* = 0.003; “Cohen’s d” = 12.32 Extremely Large Effect), and 40 min (11,808 ± 677, 95% CI 10,345.17–13,270.69, CV 21%, “e” *p* = 0.018; “Cohen’s d” = 11.12 Extremely Large Effect). There was also a noteworthy response to CA between 24 h (9884 ± 365, 95% CI 9094.36–10,692.79 CV 14%) and the moment after (12446 ± 686, 95% CI 10,964.94–13,927.37, CV 22%, “f” *p* = 0.06; “Cohen’s d” = 7.64 Extremely Large Effect), 5 min (12,209 ± 657, 95% CI 10,788.64–13,629.51, CV 20%, “g” *p* = 0.045; “Cohen’s d” = 7.55 Extremely Large Effect) and 20 min (12,367 ± 720, 95% CI 10,811.90–13,922.24, CV 22%, “h” *p* = 0.048, in Moment, η^2^p = 0.410, Large Effect; “Cohen’s d” = 6.65 Extremely Large Effect). In PL, there was a greater response between after (12,580 ± 742, 95% CI 10,975.94–14,183.77 CV 22%) and 60 min (10,233 ± 477, 95% CI 9203.47–11,263.53, CV 17%, “i” *p* = 0.004, in Moment, η^2^p = 0.410, Large Effect; “Cohen’s d” = 8.06 Extremely Large Effect). A distinguished response was also observed between the use of PL and CA at 5 min (PL 11,242 ± 511, 95% CI 10,138.61–12,345.96, CV 17%, and CA 12,209 ± 657, 95% CI 10,788.64–13,629.51, CV 20%, “*” *p* = 0.039; “Cohen’s d” = 5.17 Extremely Large Effect) and 60 min (PL 11,417 ± 957−12,878.16, CV 17%, and CA 10.233 ± 477, 95% CI 9203.47–11,263.53, CV 22%, “**” *p* = 0.027; in Interaction, η^2^p = 0.240, Medium Effect; “Cohen’s d” = 2.37 Very Large Effect). In the other moments, with the exception of 24 h, there were no differences; however, in relation to the absolute values, the use of caffeine-maintained values higher than the placebo.

Regarding estimated MVO_2_ (Figure 4B), there was a noteworthy response in exercise with CA between the moments before (7.8 ± 0.83, 95% CI 5.99–9.56 CV 40%) and after (11 ± 0.96, 95% CI 8.98–13.13, CV 32%, “a” *p* = 0.001; “Cohen’s d” = 14.85 Extremely Large Effect), 5 min (11 ± 0.92, 95% CI 8.73–12.71, CV 32%, “b” *p* = 0.003; “Cohen’s d” = 15.10 Extremely Large Effect), 10 min (11 ± 1.07, 95% CI 8.40–13.02, CV 37%, “c” *p* = 0.009; “Cohen’s d” = 9.60 Extremely Large Effect) and 20 min (11 ± 1.01, 95% CI 8.77–13.12, CV 34%, “d” *p* = 0.003; “Cohen’s d” = 12.31 Extremely Large Effect). There was an even greater response in caffeine between 24 h (7.47 ± 0.51, 95% CI 6.36–8.57 CV 26%) and after (11 ± 0.96, 95% CI 8.98–13.13, CV 32%, “e” p = 0.006; “Cohen’s d” = 7.62 Ex-tremely Large Effect), 5 min (11 ± 0.92, 95% CI 8.73–12.71, CV 32%, “f” p = 0.045; “Cohen’s d” = 7.59 Extremely Large Effect) and 20 min later (11 ± 1.01, 95% CI 8.77–13.12, CV 34%, “g” p = 0.048; in Moment, η^2^p = 0.398, Hight Effect; “Cohen’s d” = 6.67 Extremely Large Effect). In the PL group, there was a distinguished response between the before (8.00 ± 0.81, 95% CI 6.26–9.74 CV 38%) and after (11 ± 1.04, 95% CI 8.00–13.49, CV 35%, “h” *p* = 0.004; in Moment, η^2^p = 0.398, High Effect; “Cohen’s d” = 11.01 Extremely Large Effect). A greater response between PL and CA was also observed in 5 min (PL 9. ± 0.71, 95% CI 7.82–10.91, CV 29%, and CA 11 ± 0.92, 95% CI 8.73–12.71, CV 32%, “*” *p* = 0.039; “Cohen’s d” = 5.21 Extremely Large Effect) and 60 min (PL 7. ± 0.67, 95% CI 6.52–9.40, CV 31%, and CA 10 ± 0.95, 95% CI 7.57–11.66, CV 37%, “**” *p* = 0.027; in Interaction, η^2^p = 0.143, Medium Effect; “Cohen’s d” = 6.52 Extremely Large Effect). In the other moments, with the exception of 24 h, there were no differences; however, concerning the absolute values, the use of caffeine-maintained values higher than the placebo.

## 4. Discussion

The present study aimed to analyze the effects of caffeine supplementation on hemodynamic indicators in Paralympic powerlifting athletes, before and after exercise, using the 5 × 5 method at 80% of 1RM. The results showed that the consumption of 9 mg of anhydrous caffeine per kg of body weight in Paralympic did not cause a hypotensive effect, as well as in the placebo group. However, there was a greater response in 24 h with CA and PL, related to SBP, with higher absolute values for CA. Moreover, there was an increase in the 24 h response with DBP related to CA. Furthermore, when observing the absolute values before the intervention and after 24 h, related to SBP and DBP, there was a decrease in the values at 24 h for CA in relation to PL. This finding can be explained by the fact that, during physical exercise, the hydrolysis of adenosine triphosphate occurs, leading to the accumulation of adenosine in the first moments after exercise, resulting in vasodilation of the arteries by the reduction in peripheral vascular resistance [7,38]. CA competes for adenosine receptors, preventing their action [3,39]. However, the half-life of caffeine varies between 2 and 10 h [21,24,29,40], which could explain the reduction in absolute SBP values with CA only 24 h after exercise. This effect can also be explained by changes in autonomic modulation [5,23,41]. Previous findings showed an increase in parasympathetic modulation after the consumption of 3 to 6 mg kg of caffeine, in addition to suggesting that caffeine dosage and parasympathetic reactivity are not linear [23,41]. The results of this study reinforced the increase in parasympathetic reactivation 24 h post-exercise, that is, the reduction in SBP can be explained by the reduction in peripheral vascular resistance, caused by the reduction in sympathetic stimuli via the central nervous system after exercise [42]. Strength and high-intensity exercises tend to raise blood pressure [5]. In this sense, PEH may not occur, as it may vary according to the change in intensity, volume, and rest time of the exercise [4]. These factors may have contributed to the absence of a hypotensive effect in both groups. Another important point that may have contributed to these findings is the fact that the sample was composed of high-performance athletes, who tend to have fewer physiological changes [5].

In line with this, the study by Astorino et al. [3], when comparing BP changes in normotensive and prehypertensive men who completed resistance exercises after CA ingestion, found that in relation to PL, CA significantly increased SBP at rest, during and after intense strength exercise, but no change in DBP or HR, and PEH occurred in any of the treatments using higher loads of 70–80% RM. In our research, we did not check PEH, but in 24 h SBP reduction was verified in comparison to PL with loads of 80% RM. External factors may have contributed to these results, as the participants’ habits outside the laboratory were not controlled, such as sleep, food intake, and emotional stress. João et al. [5] supposed that greater exercise intensities, >80%, could lead to a reduction in cardiac output mediated by a reduction in systolic volume (SV). In a review study, Wikoff et al. [39] found mixed results with CA use, with some studies reporting small increases in BP with low CA consumption (≤100 mg/day), while others reported no effect on BP with higher CA consumption (≥400 mg/day). In our study, a reduction in BP was verified 24 h post-exercise in athletes trained using doses above 400 mg, as the average consumption of CA per athlete was 734.8 mg.

The mean blood pressure showed similar behavior for both CA and PL conditions, with a difference of only 20 min for both, where CA presented a little higher (104 mmHg) in relation to PL (100 mmHg). Apostolidis et al. [43] supplemented 6 mg/kg in trained soccer players and after 60 min of CA ingestion, the MBP means increased by 6%. In our study, even with a higher dose, the CA increased by only 2.5% after 60 min. Brothers et al. [44] investigated the impact of energy drinks (2 and 3 mg/kg CA) and found that DBP and MBP were slightly elevated, with similar results to our study. Souza et al. [42] found that the mean values at 9 h post-exercise for SBP, DBP, and MBP were significantly higher for the CA versus PL condition, in addition to a significant increase in SBP, DBP, and MBP for the CA condition at rest and PL. The results of the present study point to a tendency towards a reduction in the values of the aforementioned variables from 60 min onwards, in line with the previous study that showed an increase 9 h later. Even though we did not analyze the moment 9 h later, this trend was confirmed 24 h later.

Regarding heart rate, our study showed a statistically higher HR in the CA condition compared to the PL condition, but in both the CA and PL situations, the HR was shown to reduce after 40 min. In the literature, it has been reported that due to high metabolic demands during physical exercise, it is common for HR to increase [45]. Sarshin et al. [41] observed that increases in post-exercise HR and BP may be related to a higher exercise load and not the direct effects of CA intake. Previous research found results similar to ours, but with a reduction after 7 min, in both CA and PL protocols, with the justification of the reactivation of the parasympathetic system, in terms of physical condition and exercise intensity [38]. In contrast, the review by Wikoff et al. [39] evaluated the effect of <100 to ∼750 mg CA/day in adults, during or after exercise, and reported that most studies reported a lack of HR-related effects, and also that some other studies showed decreases in HR at exposures to CA of ≤100 mg/day, while others reported no change in HR at the consumption of CA ≥ 400 mg/day. Thus, these authors concluded that doses of 400 mg of CA/day in healthy adults would be safe concerning adverse effects on HR.

The literature has hypothesized that a reduction in HR at rest can be explained by baroreflex effects [41,46]. With the consumption of CA, BP and peripheral vascular resistance increase, and baroreceptors in the aorta and carotid arteries detect and regulate the decrease in HR to adjust BP, reinforcing the post-exercise parasympathetic reactivation [37]. In contrast, greater increases in HR during exercise have been observed in studies with a sample of middle-aged and sedentary people [46].

Regarding PPR, our findings revealed higher absolute values for the CA condition, with values increasing up to 20 min later (12367) and decreasing from 30 min (11665) onwards with constant values. There were also differences between CA and PL at 5- and 60-min post-test, where CA (12209 and 11417) presented higher values than PL (11242 and 10233). Another important point occurred 24 h later, where the CA (9883) showed a greater drop in absolute terms, even having lower values than the PL (10359). Furthermore, the PPR is the result of the multiplication of SBP by HR [47], and changes in these parameters can predict cardiovascular complications [42]. Barbosa et al. [48] found increases in PPR 45 min after consumption of CA (300 mg), as well as increases immediately after exercise, but they did not find differences in PPR between the CA and PL groups, and thus justified the finding by the fact that there was no difference in HR between the groups. The PPR is important for prescribing and monitoring physical exercise, as it is the best indirect indicator of myocardial work during exercise [49]. Thus, a pressure product rate above 30,000 mmHg.bpm would be indicative of the risk of cardiovascular problems [48,50]. In this sense, the data from this research demonstrated that the use of CA in PP would be safe concerning myocardial effort. Reinforcing this, at no time was the PPR able to reach 30,000 mmHg.bpm, with the maximum PPR reached 20 min later with CA of 12367 mmHg.bpm and with PL 12,580 mmHg.bpm soon after, which demonstrates that PP exercise would be safe in terms of cardiac overload.

In estimated MVO_2_, our study showed similar behavior to the PPR, which was already expected, since the MVO_2_ is correlated to the PPR. In this direction, with this parameter being increased, it would be indicative of a metabolic overload of the heart [2,51]. However, our findings showed increases up to 20 min later and then from 30 min later, it showed reductions in both CA and PL conditions, highlighting a large effect in relation to 5 and 60 min, showing a drop of 1.1 for CA and 1.4 for PL. According to the literature, the use of CA can cause an overload on the cardiovascular system [48]. However, as previously mentioned, MVO_2_ indicates the metabolic overload of the heart. In this sense, according to our findings, the consumption of CA in this group is safe, since there was a reduction in MVO_2_ after 30 min. The justification for these findings is probably related to the increase in HR up to 30 min and its fall from 40 min onwards, since MVO_2_ is related to PPR, with PPR being the product of SBP and HR.

Regarding the analysis of the measurement of side effects through the questionnaires, the participants measured low side effects. Among them, only four athletes mentioned increased production and elimination of urine after the test and three mentioned this increase after 24 h. CA can stimulate diuresis by inhibiting the antidiuretic hormone (ADH) [48]. In addition to these results, they measured increased activity and vigor, and improved performance. Corroborating our findings, Simões et al. [36] found that there was a low incidence of side effects in their study. Unlike our findings, Wilk et al. [31] showed an increased frequency of all adverse side effects after the ingestion of 9 mg/kg CA. Wilk et al. [52] also found a significant increase in the frequency of side effects with the ingestion of 9 and 11 mg/kg of CA. Pallarés et al. [32] found the appearance of adverse side effects with caffeine doses of 9 mg/kg.

Although some studies have shown evidence of the effects of physical activity on blood pressure [53,54], the use of supplements and caffeinated beverages on the cardiovascular response to exercise has been widely studied. Thus, Systolic Blood Pressure tends to increase during exercise with caffeine consumption [55]. In the same direction, caffeine tends to promote adverse effects, where low to moderate doses are associated with anxiety, restlessness, irritability, and nausea. Already high doses (3–5 g) can cause important physiological effects, causing palpitations, hypertension, agitation, convulsions, and even coma, with reports even of the potentiation of intracranial hemorrhage [56]. In the case of the use of coffee, several studies have shown a beneficial effect of coffee consumption, with improvements in the nervous, digestive, and cardiovascular systems, and even in the improvement of the kidney. Caffeine has bidirectional influences on blood pressure regulation, where moderate consumption does not increase and could even reduce the risk of developing high blood pressure. However, occasional caffeine consumption tends to have effects that can trigger hypertension [57]. Thus, the consumption of caffeine as energy has increased as a way to improve performance. However, the use of caffeine and energy drinks tends to increase blood pressure, tachycardia and nervousness, which can result in cardiovascular disorders [56,58]. In this sense, risks of cardiovascular diseases related to the use of caffeine have been observed, in larger amounts (5 g), or even moderate amounts (3 g) [59,60]. Thus, resistance exercise could be associated with intrinsic vessel size-dependent changes in coronary smooth muscle and endothelium-mediated regulatory mechanisms, where the proximal arteries of exercise-trained animals demonstrated greater sensitivity to the vasodilatory effects of adenosine. In this sense, the relaxation responses to adenosine is not affected by exercise, but the possible vasodilation induced by bradykinin tends to be increased, which could explain our findings [61]. Thus, our study found evidence that the use of caffeine by Paralympic powerlifting athletes was safe in cardiovascular terms.

The results of our study show that at various times there were no differences between the use of caffeine and placebo. These findings corroborate another study indicating that CA intake did not affect the recovery of heart rate variability after exercise [62]. On the other hand, the fact that the condition with the use of caffeine presented absolute values above the values with the placebo corroborates another study that indicated that the effects of caffeine on the cardiovascular system showed that this use can delay the recovery of hemodynamic indicators in the post-exercise resting state, where caffeine use was able to potentiate reductions in post-exercise parasympathetic activity, causing delayed recovery of heart rate, blood pressure, and other hemodynamic indicators after exercise [1].

However, our study had some limitations. First, the athletes’ diet was not controlled during the study period, and neither was the athletes’ sleep time before the tests. Second, we did not assess peripheral vascular resistance, sympathetic activity, stroke volume, beta-adrenergic receptors, or endothelial factors. Third, the auscultatory method was used to assess blood pressure. This method, although universally used, has some limitations in relation to invasive methods, such as intra-arterial catheterization. Fourth, the current research sample was small. Another limitation was the lack of monitoring of the athletes’ sleep, in addition, the diet was also not controlled, it was only asked to indicate the use of supplements. Finally, as the subjects studied here were all normotensive, the results cannot be extrapolated to hypertension. A strength of the study was the time of day of CA consumption and test application. Delleli et al [19] highlighted the effect of time of day on CA potential, with greater effects in the morning.

## 5. Conclusions

The literature has shown that the consumption of high doses of caffeine can lead to the risk of cardiovascular problems, as there is a suspicion of increased overload of the cardiovascular system [26,27]. In our study, we can conclude that the use of 9 mg/kg caffeine during exercise at 80% of 1RM in Paralympic powerlifting did not cause PEH. It resulted in a greater increase in SBP and DBP in relation to PL 24 h later, and promised a hypotensive effect, with lower absolute values for the CA condition in SBP and DBP. It also showed higher a HR compared to PL; however, it showed a reduction after 40 min. Although PEH did not occur, supplementation in this group can be considered safe, since it did not cause side effects and did not cause cardiovascular risk related to PPR and MVO_2_. Given the complex interactions between caffeine on metabolism and cardiovascular function in this unique athletic population, we believe that further investigations in this area should be performed.

## Figures and Tables

**Figure 1 biology-11-01843-f001:**
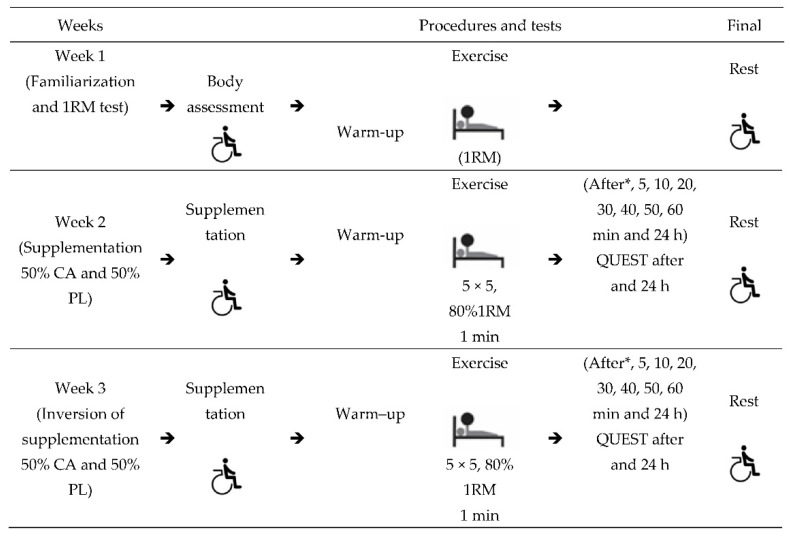
Experimental study design. * Immediately after; 5 × 5: five sets of five repetitions; 1RM: One repetition maximum; Rest: return to daily activities; QUEST: side effects questionnaire; 1 min: 1 min break between blocks.

**Figure 2 biology-11-01843-f002:**
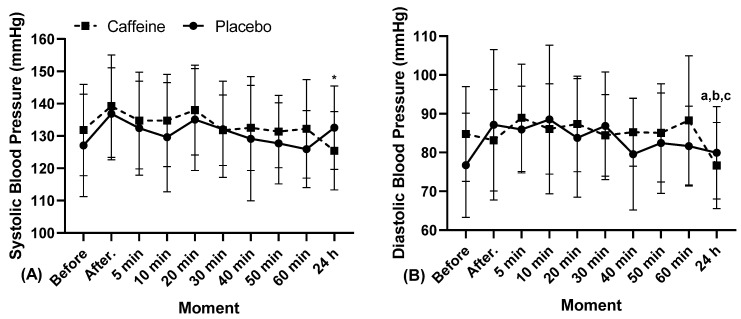
Cardiovascular response after resistance exercise with caffeine and placebo in relation to (**A**) Systolic Blood Pressure, (**B**) Diastolic Blood Pressure, min: Minutes, h: Hours. “*” indicate interclass, and “a”, “b”, and “c” indicate the intraclass difference.

**Figure 3 biology-11-01843-f003:**
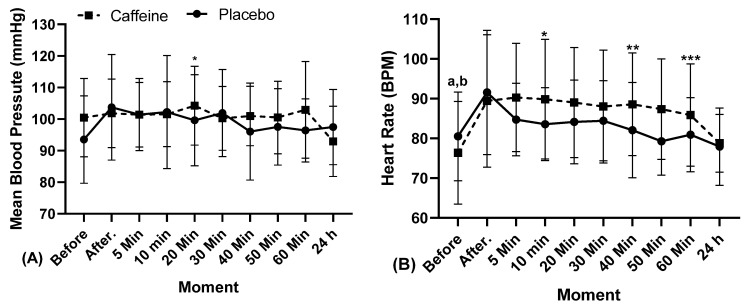
Cardiovascular response after resistance exercise with caffeine and placebo in relation to Mean Blood Pressure, Heart Rate, at different times, min: Minutes; h: Hours. “*”, “**” and “***” indicate interclass, and “a” and “b” indicate the intraclass difference.

**Figure 4 biology-11-01843-f004:**
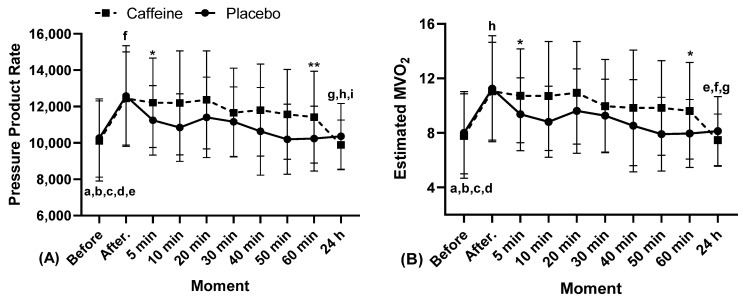
Cardiovascular response after resistance exercise with caffeine and placebo in relation to pressure product rate and estimated myocardial oxygen volume, at different times, min: Minutes, h: Hours. “*”, and “**” indicate interclass, and “a”, “b”, “c”, “d”, “e”, “f”, “g”, “h” and “i” indicate the intraclass difference.

**Table 1 biology-11-01843-t001:** Characteristics of the study participants *n* = 14.

Features	(Mean ± SD)
Age (years)	32.4 ± 8.5
Body mass (kg)	81.7 ± 21.9
Experience (years)	3.1 ± 1.0
Systolic blood pressure (mmHg)	126 ± 15
Diastolic blood pressure (mmHg)	75 ± 12
1RM test (bench press) (kg)	126.9 ± 41.2
1RM test/body mass (kg)	1.6 ± 0.4

1RM: one repetition maximum.

**Table 2 biology-11-01843-t002:** All side effects and number (frequency) of participants who reported side effects immediately after the test protocol with CA (Questionnaire of Side Effects (QUEST) + 0 h) and 24 h after (QUEST + 24 h).

Side Effects	After	24 H
Headache	0 (0%)	3 (21%)
Abdominal/gastrointestinal pain	0 (0%)	0 (0%)
Muscle pain	6 (43%)	5 (36%)
Increased energy and strength	11 (79%)	10 (71%)
Tachycardia/palpitations	1 (7%)	0 (0%)
Insomnia	1 (7%)	1 (7%)
Performance improvement	13 (93%)	9 (64%)
Increased production and elimination of urine	4 (29%)	3 (21%)
Increased worry/anxiety	0 (0%)	1 (7%)

Data are presented as the number of participants (frequency) who responded affirmatively to the existence of a side effect.

## Data Availability

The data that support this study can be obtained from the address: www.ufs.br/Department of Physical Education, accessed on 11 June 2022.

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
