# Peer review of "Does Caffeine Supplementation Associated with Paralympic Powerlifting Training Interfere with Hemodynamic Indicators?"

_biology, 2022, doi:10.3390/biology11121843_

Round 1
Reviewer 1 Report (New Reviewer)
This study involves the effects of caffeine supplementation on spinal cord disabled (SCD) power lifters. Although the study has general merit and interest to the disabled athlete sports science community, I believe that the introduction and thrust of the study description does not clearly link the importance of understanding caffeine supplementation and any negative hemodynamic responses prior, during and after resistance exercise directly to this special population - in other words, why is this important to the SCD athlete? - what makes this population at risk for adverse effects of caffeine- the nature (i.e., hypertensive response) of upper body resistance exercise? Alternatively, what benefits of caffeine are sought in this population? increased neuromuscular efficiency or metabolic adaptions ? please elaborate.
What are potential inherent cardiovascular regulatory mechanisms in play with SCD athletes during resistance exercise? , and /or their risk of developing future cardiovascular disease?
The study utilized male athletes only: why are females not studied - this should be mentioned in the methodology.
How do the authors control for habitual caffeine consumption? it would appear that a better experimental design would be to include a 1 week wash- out period so that all study volunteers are at a blood caffeine baseline level and between cross-over periods.
Please clarify the measurements utilized to measure hemodynamic responses: was an automated BP device used for all measures? Understand the SBP, DBP and MAP measures, but I am confused regarding the estimation of myocardial oxygen consumption (MVO2) (Section 2.3.2): This is not usually referred to as a 'volume' . How is minute ventilation (VE) (lines 164 - 165) relate to the estimation of MVO2 ? If VE was measured for input into this equation, how was it measured or estimated? Why not simply estimate myocardial oxygen demands using rate pressure product (SBP x HR)? In some instances, "DP" is introduced -- is this interchangable with RPP in this study?
More explanation on the chosen dose of 9 ml/kg caffeine is needed and backed up by the literature for this unique population.
For the results, the inclusion of all the CI data in text is distracting and redundant- please delete from the text and summarize the statistics. The significance letters can be explained in the figure captions. The y axis label for Figure 3a should be 'Mean Arterial Blood Pressure" vs. 'Medium'. Figure 4: please label the axes with terms consistent with those outlined in the methods in section 2.3.2. For example, in section 2.3.2, DP is mentioned but not defined (which should be SBP x HR) , then it is presented in Figure 4a. In figure 4b, please include the units of MVO2 on the y axis, and more appropriately, term this as 'MVO2 estimated'.
In the Discussion I would like to have seen a more clear analysis of how these results apply to the specific population studied (SCD) power athletes. How might post-exercise hypotension result in negative immediate and/or chronic reactions and impairment of performance as well as produce long term health implications? please relate other studies using this same population, vs. comparisons to non-disabled adults and non-disabled athletes. What are the unique hemodynamic responses to maximal resistance exercise (acute rise in BP and peripheral resistance for example), that occurs which place great loads on the myocardium, that in combination with caffeine ingestion, might place these athletes at risk?
On page 10, lines 362-368, several numerals appear in the text; what are these? I suspect blood pressure values, but perhaps missing the decimal? please clarify..
The Conclusions should also summarize these thoughts clearly. For example, the sentence on line 416 " during training at 80% 1RM in paralympic powerlifting can be interesting"... this statement is vague and does not clarify the take home message of the findings of this study.
Author Response
Please see the attachment.

Reviewer 2 Report (New Reviewer)
The title needs to be revised as the word “associated” seems to suggest there is always a relationship between caffeine and power lifting.
The abstract is not clear on the duration of the training. Please clarify throughout the manuscript. If the exercise was always the same, then there was no progression, so was this really then a physical training programme.
Ls 45. The conclusions need to be revised and be specific to what was done and observed in the study.
Throughout the manuscript, it needs to be made clear whether the changes, e.g. in blood pressure were meaningful and have really practical application.
Please provide the coefficient of variation of the measurements.
The introduction needs to get substantial revision to clarify that the focus is on post-exercise observations.
L86. Do you mean studies are inconsistent with observations? Is controversial the right term?
Ls 92-97. Please clarify under what conditions the effects of caffeine was examined.
L106. Please clarify what is meant by “supine movement”.
Fig. 1. How was the randomization done as exactly the same number of participants were starting in the placebo or supplement condition.
Fig. 1. Change “Teste”.
Fig.1 Change “BP e HR”.
L128. Do we need all of these years?
Table 1. SD of 21.2 different than in L114.
Table 1. Please express blood pressure without decimal places throughout the manuscript.
Table 1. Please delete “Legend:”.
L145. Please delete “BP” if it means blood pressure.
L160. I suggest to replace “cardiac pressure product” with “rate pressure product” and change the abbreviation “DP” to “RPP”.
L182. Please delete “in the morning (”.
Figures 2 and 3. Please delete “Legend:”.
Figure 3. Change “medium” to “mean”.
Figure 3A. The SDs at 40 min are very different than at other time points. Why is that? Please provide a comment in the manuscript.
Throughout the manuscript, please provide heart rate without decimal places.
Figure 4 needs a legend on symbols in the figure.
Ls 403-410. Are these all limitations of the study. It needs to be make for example why dietary intake may have had an effect. Please revise the reconsider the content of this section.
There is inconsistency in the references style. Please consult author guidelines.
Author Response
Please see the attachment.

Reviewer 3 Report (New Reviewer)
The present study analyzed the effects of caffeine supplementation on hemodynamic indicators in Paralympic powerlifting.
The authors concluded that the use of 9mg/kg of caffeine during training at 80% of 1RM in Paralympic powerlifting can be interesting.
The manuscript is well written and present interesting findings. However, some modifications are required:
I suggest that some recent studies are helpful for reinforcing the introduction and discussion:
Delleli et al. (2022). Nutrients, 14(14), 2996.
I suggest discussing the limitations. How these proposed limitations could affect the results according to the literature.
Also, I suggest discussing limitation or adding information about the sleep of participants and the time of day of the experimentation.
A practical application part could be helpful.
Author Response
Please see the attachment.

Reviewer 4 Report (New Reviewer)
This study aims to evaluate the effect of caffeine supplementation on post-exercise cardiovascular response in Paralympic weightlifters. The authors conducted an interesting work; however, the manuscript needs to be improved and the interpretation of the results needs to be better interpreted. Some suggestions and comments to improve the document are presented below:
Introduction.
The introduction presents the ideas in a fragmentary way and without a defined line of argument. As a suggestion, I suggest to the authors structure the introduction into 3-4 paragraphs to identify in the first paragraph the context of the research (cardiovascular response after resistance exercise). In a second paragraph, introduce the independent variable (caffeine use) and the relevant physiological information and performance implications of its use (among other aspects). Finally, to briefly develop the evidence to date on the use of this supplement in resistance exercise (both CP and PP), which will lead to the rationale for this study - caffeine use in paralympic athletes -, the objective and the research hypothesis.
In this section, it is important to differentiate acute responses from chronic adaptations of resistance training on the cardiovascular system and to provide references for each of them. Authors are encouraged to provide direct citations (and if possible, to reference articles in the field, i.e., ESC, AHA, ACSM statements...).
2. Methodology
2.1. Research design
It is suggested to change "research design" to "study design".
It is better to talk about sessions and not weeks to understand and not misunderstand the study.
Line 105. The authors should explain what the participants were doing during the "rest" of each of the sessions.
Line 106: change "supine movement" to "bench press exercise".
Line 107: The protocol should be accurately described despite being referenced to a previous work.
Figure 1. It is suggested to remove the ASS section. This is mandatory and common to all studies. Correct the legend for 5x5 (five sessions change it to five sets). Introduce rest time between sets. Explain what rest means also in the figure.
2.2. Participants
If the number is expressed in the text, omit the parenthesis with n=14. Since a table with the characteristics of study participants is introduced, remove this information from line 114. As a suggestion in the first sentence, I would only explain the number of participants. Thus, in the following sentences of the paragraph, only the characteristics of the participants (project, ranking, etc.), inclusion criteria (18 months of experience, prerequisites, etc.) and exclusion criteria are described.
If informed consent is explained in this paragraph, delete it from paragraph 2.1.
2.3. Instruments
Some errors have been detected in this section. Revise the capacity of the scale (300 or 3000 kg), the references of the instruments and the wording. I would suggest rephrasing the sentences in this section as follows (or similar) to make the sentences less fragmented and for better understanding: Body mass was measured seated on an electronic wheeled scale (Micheletti, São Paulo, Brazil), of the digital electronic platform type, with a maximum weight capacity of 3000 kg (dimensions 1.50 m by 1.50 m). To perform the bench press exercise, an official 2.10 m long straight bench (Eleiko, Halmstad, Sweden) and an Olympic barbell (220 cm total length, weight 20 kg), both approved by the International Paralympic Committee [13], were used. BP and HR were assessed using an automatic non-invasive blood pressure monitor (3AC1-1PC, Microlife, Widnau, 146 Switzerland).
Line 142. Delete reference in another style not allowed.
Line 147. Provide an appropriate validation reference (e.g., 10.1097/00126097-200512000-00008).
Line 149. Remove the parenthesis with anhydrous, even though the abbreviation includes this term.
Line 158. The cardiovascular measuring device allows measurement of SBP, DBP and HR, while the rest of the variables must be calculated. Appropriate use of the terms, to measure, to assess and to record is important and should be reflected in the manuscript.
Line 159. A single term should be used for the double product or explain which terms will be used indistinctively.
Line 162. Explanation of the calculation of MVO2 more concise and reference to the formula is more appropriate.
Line 166. Information is repeated on line 161. Please, the authors should revise the manuscript to provide a document as synthesised as possible, allowing the study to be replicated and with references to the reliability and validity of the devices to support the quality of the study.
2.3. Load Determination
This section should appear at the beginning of the methodology. It should provide a reference to the 1RM protocol performed and in addition (as there is no word limit) explain the 1RM procedure step by step.
2.3.4. Side effects
The side effects registration is remarkably interesting; however, it is not reflected in figure 1. Authors should indicate the list of nine questions or symptoms they registered and if they were assessed in the same way both times.
2.4. Procedures
Most of the information mentioned here has already been previously explained. To improve the understanding of the methodology, following an order and to avoid repetition of information, it is suggested to reorganise the Material and method section into the following sub-sections: Study design, participants, procedures (indicating the 3 sessions and the procedure to be followed in each one and the characteristics of caffeine supplementation), data recording analysis (cardiovascular and side effects) and statistical analysis.
On the other hand, it is important to emphasise the requirements that participants must consider after the session, as a measure of BP is recorded at 24 h. These requirements must be described in the text (e.g., no exercise, substance use, etc.).
2.5. Statistical analysis
Line 212: Indicate which tests would be performed to compare side effects between experimental sessions (AC and PL), if applicable.
Line 212-213: "To assess performance between groups?" This work aims to examine the cardiovascular response after exercise using a repeated measures design.
Please, indicate the number (2x2) and the factors in the ANOVA.
Line 214: In addition to the effect size of each factor, the effect sizes of the pairwise comparison should be included.
3. Results
Throughout the manuscript, there are no consistent abbreviations for the placebo session. Review the use of abbreviations.
Line 220: Please explain what is meant by "increased activity".
Table 2. It is not detailed from which session the data are reflected (CA or PL).
Line 228: A paragraph consists of at least three sentences. Join this sentence to the following paragraph.
Figures 2, 3 and 4. It is suggested to merge all figures into one figure and change the title of the figure to "Cardiovascular response after resistance exercise with the use of caffeine and placebo" or a similar sentence. Symbol meanings should be explained in the figure legend and never appear in the text. It is suggested, as the journal accepts, to use colours to differentiate the sessions between them and their pairwise comparison symbols and to put in only one direction for the error bars.
Line 235, 238, 249, .... References in the text of the figures should be set out as indicated in the journal.
Line 235. Which statistics do the confidence intervals refer to?
For each variable, report the results of the ANOVA (main effect of each factor, interaction and, if there is an interaction between factors, pairwise significant comparisons, and their effect sizes - both for within-session and between-session comparisons). The wording of the results must be carefully edited, being consistent with the terms that define the interventions, the variables, etc.
4. Discussion
It is important to be careful with the drafting. The hypotensive effect is caused by training, and this may be potentiated/inhibited by the use of supplements such as caffeine. Also, the data provided do not allow us to describe that such an HPE exists, as no significant differences are reported between pre-exercise and post-exercise values.
Since descriptive data are reflected in the text in the results, they should not be introduced in the discussion. As a practical alternative, the total reduction for that variable could be shown.
The authors provide physiological reasons to justify the results obtained. Could external factors (e.g., habits of the participants outside the laboratory) justify that differences only exist after 24 hours? Have these been controlled for?
Line 323. Refer to the original work.
The discussion should be improved. Results from previous work should not be reported and explained in the discussion. They should only be confronted with the findings to identify justifications for the results. For conclusion, it is recommended to be more synthesised and to focus on the results obtained and their direct applications.
Minor revisions.
Appropriate use of abbreviations. If they are not used repeatedly, omit their introduction. Sometimes the same abbreviation is introduced twice.
Line 485: Reference 18.
Round 2
Reviewer 1 Report (New Reviewer)
Many of the original issues with the manuscript have largely been dealt with in this revision, and it reads in a more clear manner.
Just a few minor additional suggestions:
1. line 49. please consider a re-phrase in the abstract. eg., : This study was performed within three weeks.
2. line 186. please define 'SD' in the MVO2 equation
3. Line 484. replace with '... it resulted in a..
4. line 488. The conclusion still needs a little more work for readability. Please correct English grammar. The last line could be improved: eg., replace "However, it needs more investigations" with (example): "Given the complex interactions between caffeine on metabolism and cardiovascular function in this unique athletic population, we believe that further investigations in this area should be performed }.
Author Response
Please see the attachment.

Reviewer 2 Report (New Reviewer)
According to the response letter, some suggested changes were actually not made. Therefore, here is again an extensive list.
Throughout the manuscript, it seems the words “exercise” and “training” are used interchangeably and that is incorrect. Training involves a program of overload and progression over a period of weeks or months. Use the word “exercise” when you communicate about acute and short-lasting responses and use the term “training” when you communicate about adaptations that occurred over weeks or months.
I suggest to replace the word “training” in the title with “exercise.
Here a few example as well where words need to change. It is the authors responsibility to check carefully were changes are needed.
e.g. L38. I suggest to replace “exercise” with “training” as it is a training programme that can have beneficial adaptations.
e.g L44. “were trained” should be “performed the exercise”.
L88. Change “. [16,17,18].” to “[16,17,18]”.
Ls 89 and 90. Please clarify that it is mg/ kg body weight.
Figure 1. Change “Teste”.
Figure 1 Legends had duplication “Rest: return to daily activities” 2x?
Figure 1. Please delete “Legends”.
Table 1. Please express blood pressure without decimal places throughout the manuscript.
Table 2. Please delete “Legend:”.
L249 and L265. Unclear what is meant by the “kinetics of training”. Please revise.
Please express blood pressure, heart rate and RRP without decimal places throughout the manuscript.
Figures 3 and 4. Please delete “Legends”.
L477. Please reconsider “erogeneity” due to meaning of the word.
There is inconsistency in the references style. Please consult author guidelines.
Author Response
Please see the attachment.

Reviewer 3 Report (New Reviewer)
I suggest that this version is suitable for publication in the journal "Biology".
Author Response
Dear Reviewer,
thank you very much for your help in improving the manuscript!
The authors
Reviewer 4 Report (New Reviewer)
After reviewing the comments of my peers and the authors' responses, I consider that the article still has considerable flaws that need to be addressed.
In general, the authors should improve the drafting of the manuscript.
The study aims to evaluate the effect of caffeine consumption in PP athletes. Therefore, the introduction should provide a literature-based rationale for the study's value and relevance. In broad terms, the introduction should highlight at least two important points. First, the cardiovascular response to exercise in healthy individuals and describe what factors are modified in PP athletes (e.g., alterations in autonomic regulation). This would justify the relevance of the study. If there were no differences in cardiovascular response between CP and PP, why should they be studied? Secondly, the authors should describe what differences are produced by caffeine consumption in the cardiovascular response and the existing evidence in the population they are studying. It is recommended that the authors review all the comments provided by the review panel and rewrite the introduction, adding all the listed points not addressed in the first review, as well as those added in the second review.
It is suggested that the English be proofread by an expert or native speaker. In some paragraphs, it is difficult to understand the information for this reason.
When describing the results, it is suggested not to use phrases such as "there was a statistical difference". Instead, describe which situation (CA or PL) provides the greater or lesser response. The description of the results is difficult to read and interpret. Again, I suggest that the information related to the meaning of the symbols in the graphs should not be included in the text. Instead, it should be explained in the legend of the figures or tables (see the information for authors and other published articles to see how this is usually done).
Table 2 does not indicate to which session the side effects correspond. Although the authors do not attempt a statistical analysis of side effects, it is important that the information reported is concise and cannot be misinterpreted.
Concerning the analysis of the results, observing the graphs, and considering the current evidence, it seems that some results have been omitted, for example between the values obtained before and after exercise. If this is not the case (i.e., if no differences were observed between the values recorded before and just after the exercise) it is important to explain it, as this would be an important argument for the possible responses obtained. A more comprehensive interpretation of the results is needed, regardless of whether there are differences between protocols.
The discussion (as well as the introduction) needs to be improved. An important finding is that there is no significant reduction in blood pressure (hypotensive exercise effect) in any of the sessions. It should justify what factors may be contributing to this, and then explain the differences between sessions and the possible reasons for these differences.
The rationale for some of the results should be improved. The physiological reasons given must be interpreted and related to the results obtained. It cannot be a simple enumeration of findings and citations.
In conclusion, the article must be proofread in its entirety, attending to each of the reviewers' observations and especially to the English drafting and use of the correct terms (e.g., exercise and training, pressure indicators, mentioning blood pressure response when all other variables are also mentioned, etc.).
Attention should also be paid to minor issues such as punctuation, capitalisation, and the use of abbreviations (e.g., CA or AC). After reviewing the comments of my peers and the authors' responses, I consider that the article still has considerable flaws that need to be addressed.
In general, the authors should improve the drafting of the manuscript.
The study aims to evaluate the effect of caffeine consumption in PP athletes. Therefore, the introduction should provide a literature-based rationale for the study's value and relevance. In broad terms, the introduction should highlight at least two important points. First, the cardiovascular response to exercise in healthy individuals and describe what factors are modified in PP athletes (e.g., alterations in autonomic regulation). This would justify the relevance of the study. If there were no differences in cardiovascular response between CP and PP, why should they be studied? Secondly, the authors should describe what differences are produced by caffeine consumption in the cardiovascular response and the existing evidence in the population they are studying. It is recommended that the authors review all the comments provided by the review panel and rewrite the introduction, adding all the listed points not addressed in the first review, as well as those added in the second review.
It is suggested that the English be proofread by an expert or native speaker. In some paragraphs, it is difficult to understand the information for this reason.
When describing the results, it is suggested not to use phrases such as "there was a statistical difference". Instead, describe which situation (CA or PL) provides the greater or lesser response. The description of the results is difficult to read and interpret. Again, I suggest that the information related to the meaning of the symbols in the graphs should not be included in the text. Instead, it should be explained in the legend of the figures or tables (see the information for authors and other published articles to see how this is usually done).
Table 2 does not indicate to which session the side effects correspond. Although the authors do not attempt a statistical analysis of side effects, it is important that the information reported is concise and cannot be misinterpreted.
Concerning the analysis of the results, observing the graphs, and considering the current evidence, it seems that some results have been omitted, for example between the values obtained before and after exercise. If this is not the case (i.e., if no differences were observed between the values recorded before and just after the exercise) it is important to explain it, as this would be an important argument for the possible responses obtained. A more comprehensive interpretation of the results is needed, regardless of whether there are differences between protocols.
The discussion (as well as the introduction) needs to be improved. An important finding is that there is no significant reduction in blood pressure (hypotensive exercise effect) in any of the sessions. It should justify what factors may be contributing to this, and then explain the differences between sessions and the possible reasons for these differences.
The rationale for some of the results should be improved. The physiological reasons given must be interpreted and related to the results obtained. It cannot be a simple enumeration of findings and citations.
In conclusion, the article must be proofread in its entirety, attending to each of the reviewers' observations and especially to the English drafting and use of the correct terms (e.g., exercise and training, pressure indicators, mentioning blood pressure response when all other variables are also mentioned, etc.).
Attention should also be paid to minor issues such as punctuation, capitalisation, and the use of abbreviations (e.g., CA or AC).
Round 3
Reviewer 2 Report (New Reviewer)
Please express blood pressure, heart rate and RRP without decimal places throughout the manuscript.
Author Response
Please see the attachment.

Reviewer 4 Report (New Reviewer)
The authors present a very interesting study, but the drafting remains deficient. Some general changes and additions are suggested. Please read the recommendations carefully and if any questions arise, do not hesitate to address them in the response to the reviewers.
- Check punctuation, capitalization and spaces between words (e.g., line 4, 6).
- English needs to be improved. Several expressions are literal translations from Portuguese or other Latin language where grammatical norms are completely different (e.g., line 41: the correct expression would be: ...during bench press exercise.). Excessive use of subordinate sentences. English (and especially scientific writing) avoids this use by employing simpler sentences. Authors are encouraged to make the effort to synthesise their manuscript to improve its quality. Some comments are shown below.
- Many statements lack a supporting reference (e.g., line 98). If there is no evidence in the literature, it is an argument that reinforces the need for this study.
Line 2: I would change the title to an affirmative one and modify the term "haemodynamic indicator" to "post-exercise haemodynamic response". This would make clear the assessment timing and the variables analysed.
Line 44. As indicated in previous reviews, it is important to differentiate between training (chronic, long-term effects) and session or exercises (acute, short-term effects). When referring to beneficial effects of exercise (i.e., reductions in all variables assessed in this paper) these refer to a training intervention (long-term). Authors are encouraged to introduce the idea of the benefits of resistance training in the introduction and to focus on acute responses in the abstract, as this is the aim of their study.
Line 63. For the description of the hemodynamic behaviour during and after exercise, more appropriate references should be used. If normal hemodynamic response (not using supplementation) is being discussed, it is recommended to refer to works on this topic. For example, Cornelissen et al. or Cassonato et al. have done some review articles. Another important aspect is the difference between physical exercise and resistance training. Although some aspects are similar, resistance exercise has its own unique characteristics, which are quite different from other forms of physical exercise such as aerobic exercise, for example.
Line 73. More appropriate references could be used, for example:
- Brook RD, Appel LJ, Rubenfire M, Ogedegbe G, Bisognano JD, Elliott WJ, et al. Beyond Medications and diet: alternative approaches to lowering blood pressure. Hypertension. 2013 Jun;61(6):1360–83.
- Arnett DK, Blumenthal RS, Albert MA, Buroker AB, Goldberger ZD, Hahn EJ, et al. 2019 ACC/AHA Guideline on the Primary Prevention of Cardiovascular Disease: A Report of the American College of Cardiology/American Heart Association Task Force on Clinical Practice Guidelines. Circulation. 2019 Sep;140(11):e596–646.
- Pedersen BK, Saltin B. Exercise as medicine - Evidence for prescribing exercise as therapy in 26 different chronic diseases. Scand J Med Sci Sport. 2015 Dec;25:1–72
Line 83. After "bench press" authors must insert the word "exercise".
Line 85. Replace the verb "make use" with a more appropriate verb: consume, use, etc.
Line 75. Since this line introduces the rationale for why it is important to study Paralympic athletes, the reviewers suggest that it should insert here the notion of the athlete PP and the modality:
…lower limbs [12]. In this regard, paralympic powerlifting (PP) is a modalidy adpted from …;
followed by the idea of caffeine use, and then describing that its use can interfere with haemodynamic response (line 90):
… However, it can interfere with BP and HR [19,20,21], leading to concerns about its use since people with lower limb disabilities tend to have more risk factors for high blood pressure [13]. This is especially relevant for Paralympic athletes, as 12% of them have cardiovascular abnormalities and 2% have a high risk of sudden cardiac death, in addition to other cardiac diseases [4].
This paragraph is an example (it is not required to be written in this way) that reflects the synthesised line of argument that the reviewers have requested in previous revisions.
Line 123. The wording of the sentence suggests that in both sessions the participants were supplemented and this is not the case.
Line 124. When referring to bench press it should be specified if it is the exercise, mauqina. Rephrase the sentence:
…In both sessions, five sets of five repetitions (5x5) with the 80% 1RM load and 1 min of rest between sets was perfomed for the bench press exercise, as previously described by Austin and Mann [26]…
134. As previously reviewed, the "n=14" should be removed.
134. As previously mentioned, the use of subordinate sentences should be avoided.
157. include here the n=14 (e.g. next to Characteristics of the study participants).
183. The reviewer suggest introducing this sentence "The protocol proposed by Fleck and Kraemer was used [31]" before the test description.
213. Replace draw by randomisation. and soon after by Subsequently.
215-216. Not for this study, but in the future, it is suggested that a baseline measurement of BP and HR after 50 minutes of rest also be collected.
253. Insert a space between the table and the paragraph.
Figure 2. Abbreviations are not necessary.
260. The authors again omit the anova results for each factor and the interaction between them. The post hoc results alone are not sufficient. I would again point out that the meaning of the symbols should be included in the legend and not in the text.
η2p should be consistently either with or without subscript. Also, as it is worded, it is not possible to determine which factor the effect size of the η2p refers to. For each variable should be reported: Main effect of the moment factor and its η2p, main effect of the condition factor and subsequently, report whether there is interaction or not. If interaction between factors is found, post hoc analyses should be reported for all comparisons: For each moment, inter-session comparison (CA vs. PL), and for each condition, inter-moment comparison (Pre vs. After, 5,10, ...).
345. Add athletes.
348. The sentence is unclear, it should be rephrased. Anyway, the statement made is not correct based on the results reported in the graphs. Please note that the anova analysis should be include with all the previously described parts (main effects, interactions and post hocs). Additionally, authors are encouraged to review the use of the terms increase/decrease, higher/lower, etc. to refer to the hemodynamic response. The former (and synonyms) should be used for intra-session comparisons while the latter should be used for inter-session comparisons, not vice versa.
354-362. The argumentation needs to be revised. It becomes difficult to follow the argumentation.
363-365. PEH is a reduction of blood pressure after exercise compared to previous values, not an increase. I reiterate the importance of revising and rewriting the article.
365. The existing literature on the hypotensive effect of resistance training has recently been meta-analysed in a paper by Casonatto et al 2016. Although intensity is a determining factor of PEH, in this protocol (80% 1RM) the load was enough to promote PEH, and may be the total volume performed (only 25 repetitions) and the rest time the factors that determine the absence of PEH.
Idem at 348.
375. This is a limitation
Some inconsistencies in the discussion have been detected. As previously described, certain terms are used in the wrong way, which impedes the understanding of the results. Also, it seems that the authors confuse the fact that after 24h there are differences between sessions with a possible reduction in BP.
The study, despite its interesting and novel idea, has major shortcomings in the manuscript's writing. It is recommended to perform a more comprehensive literature review (e.g. the works of Cornelisen et al., Casonatto et al. can help to improve the introduction and interpretation of the results). The authors are strongly encouraged to rewrite the article and resubmit it to the journal, as making this effort will improve the understanding of the cardiovascular response to resistance exercise in this specific population, which is less studied due to the complexity of sample recruitment.
Round 4
Reviewer 4 Report (New Reviewer)
To the authors:
The reviewer really appreciates the work done by the researchers in implementing changes and addressing comments. However, there is a major concern regarding the statistical analysis report and its interpretation.
Again it is suggested that the anova should be presented in its entirety. As the results are shown it appears that for some variables there may be no interaction between factors, so a post hoc comparison should not be performed. Also, if such an interaction exists it should be reported as well as the main effects of each factor with its partial eta squared (in this case, the authors confound the effect sizes of the main effects and the pairwise analysis).
Please review this issue, as it directly affects the interpretation of the results. For example, if there is no main effect of time this suggests that regardless of the protocol there are no alterations in the variables. The interpretation of the differences between protocols should be considered but not misinterpreted as evolutions within the session.
This is a well-conducted study that can add valuable information to the scientific evidence, but the authors need to be prudent about reporting and interpreting the data.
Round 5
Reviewer 4 Report (New Reviewer)
Authors have carried out a very interesting study, with a simple design and it addresses a gap in the literature. Moreover, the sample selected is not commonly assessed, thus increasing the potential publishability of their work.
Unfortunately, the statistical analysis and the derived results are not correctly reported, which is detrimental to the study. This has been commented in each and every one of the reviews I have done (and other reviewers have also detailed it), however the authors have not corrected this error.
The results section should include paragraphs with the following structure: Regarding XX (variable, i.e. SBP for example), main effect of condition (p = x.xxx; pη2 = x.xxx), moment (p = x.xxx; pη2 = x.xxx), and a condition by moment interaction were detected (p = x.xxx; pη2 = x.xxx). For pairwise comparison, ....(enter significant contrasts ONLY if an interaction between factors is observed. For each pairwise comparison, report mean values and standard deviation, p-value and cohen's d). I remain at your disposal for any questions you may have.
Author Response
Dear reviewer,
thank you very much for your efforts and patience to review this manuscript.
On the manuscript, you can see the changes highlighted in yellow in the Results section. The main effects of Condition and Moment and the interaction effect have been specified.
I hope this version of the manuscript is fine.
Best regards,
The Authors
Round 6
Reviewer 4 Report (New Reviewer)
Después de revisar el manuscrito seis veces y hacer observaciones, los autores no han introducido los cambios y sugerencias necesarios.
Como se ha dicho en las revisiones anteriores, el estudio es bueno, pero hay que mejorar el análisis y el informe de resultados, ya que el hecho de no informar y considerar cierta información del anova condiciona la interpretación de los resultados.
Como ilustran las figuras, el efecto del tiempo y su interacción con la condición parece no ser siempre significativo, lo que significa que los resultados mostrados pueden ser estadísticamente incorrectos. Informar del tamaño del efecto y la significación de cada factor y su interacción para todas las variables podría resolver este problema. Y esto podría hacerse en el texto (como se sugiere) o en las figuras o en una tabla.
Animo a los autores a reconsiderar la realización del análisis de nuevo, teniendo en cuenta las cuestiones estadísticas necesarias.
This manuscript is a resubmission of an earlier submission. The following is a list of the peer review reports and author responses from that submission.
Round 1
Reviewer 1 Report
The manuscript of Menezes and colleagues evaluated the possible side effects of caffeine supplementation on hemodynamic variables in paralympic powerlifters. The study is correctly designed as a randomized crossover control trial. However, the measurements are limited to blood pressure and heart rate evaluatio,n and some of the parameters, specifically MVO2, are poorly described and justified. The manuscript also needs extensive editing of the English language.
Reviewer 2 Report
The present study aimed to analyze the effects of the influence of caffeine supplementation on hemodynamic indicators in Paralympic powerlifting, before and after training, using the 5x5 method at 80% of 1RM. The results conclude that the use of 9mg/kg caffeine during training at 80% of 1RM in Paralympic powerlifting may be of interest.
The study is interesting although it is necessary to explain the results better, even in more detail and to make again an analysis of these results and conclusions.
Abstract
Line 39. Set meaning of acronym MVO2.
Introduction
Line 58 Set meaning of acronym BP.
I note that the acronym BP is set in line 86. BP appears again in line 143. DBP is described in line 143, but appears at the beginning of the manuscript (line 57). In addition, the acronym CA is indicated on line 76 and then on line 100. Please check all acronyms in the text and put their meaning the first time and only once in the manuscript.
Check the rest of the acronyms.
Materials and Methods
In figure 1, I would specify that 50% of the participants ingested CA and 50% PL and the following week it was the other way around, 50% PL and 50% CA.
Line 154. From the time you ingested the capsules until you started the test, how long did it take?
The meaning of acronyms is repeated. Please check. Example, DBP line 143, line 157,
In the abstract called DP to the double product. In line 160 you call it HPP? Is it the same thing? Correct.
Line, 162, What is MVO2? this is the acronym. Not clear, explain better.
Line 169, “In it, each participant started the trials with a weight that they could lift with maximum effort only once”. The test started by lifting a load with maximum effort? Then weight was added until it was lifted only once? “Later, more weight was added until it reached the maximum load that could be lifted at once”. Explain further.
Add the warm-up performed.
Line 213, Two-factor ANOVA, one of which was with repeated measures? specify.
Line 214 p<0.05 p italicise throughout the document.
Results
I think they should present the results better and extract a better analysis from them (Set confidence intervals? Tables? Means and standard deviations? Percentages?
In figure 2, clarify in the legend the meaning of a, b, c and the asterisk. Do the same for the rest of the legends in figures 3 and 4.
Line 241, are there no significant differences between the 24-hour and 60-minute time points in caffeine training?
I find it strange that the value in the 60 minute is higher than in the 50 or 40 minute and yet there is no difference. The same is happening with the 5 minute post-exercise DBP in caffeine training. No significant difference either? Clarify.
Also, there are no significant differences between the different times in DBP in placebo training? You need to include all results for both SBP behaviour, DBP, etc. for both caffeine training and placebo training in all figures.
Reference the figures in the manuscript.
In figure 3 B, no difference is found between caffeine training and placebo at minute 50?
Discussion
Lines 277-282. Urine production and elimination were measured and compared with Caffeine and Placebo?
If this was not measured, it is not possible to say that there was an increase in urine production an elimination after the test or a greater increase in strength.
Line 281-282.
Was strength in training measured and compared with caffeine and placebo?
If this was not measured, it is not possible to verified an increase in activity and force and a performance improvement.
If the same 5 x 5 test was done at 80% of 1RM, it is not possible to know through a questionnaire if there was an improvement in performance, because they did the same protocol. Here hemodynamic variables were measured and it is not possible to know if there was an improvement in performance because they did the same series and the same repetitions. I do not think this is the main objective to put it at the beginning of the discussion.
Lines 288-289, The results indicated that only significant differences are found with caffeine training with respect to placebo in SBP. Not in DBP.
Lines 290-296. Furthermore, in the results of the study, no differences are observed between SBP and DBP in caffeine training with respect to placebo. Only in Medium BP, HR min 10, 40 and 60, in DP and MVO2 at min 5 and 60.
Therefore, that "CA competes for adenosine receptors, preventing their action [1,37]. However, the half-life of caffeine varies between 2 and 10 hours [15,22,33,38], which would explain the reduction in blood pressure indicators 24 hours after exercise, should be clarified, since there is not much difference from minute 1 to 60, nor in DBP 24 hours later.
Line 329, The interpretation of the results I make is that at min 40 there are significant differences and also at minute 60. And at minute 50 is where the HR value is lowest when supplemented with PL, (I don't understand why there is no significant difference). So, HR was reduced with CA, but even more with PL.
Line 350-351. These values are not significant and are very similar. In any case, since no numerical data are given in tables or in the text, it is not possible to evaluate the exact absolute value in order to be able to assess correctly. In my opinion I would put the results in more detail, even with confidence intervals, statistical power and even percentages. I think the interpretation of the results would be clearer for the reader.
Line 378-384. In the limitations section, the sample of subjects could be larger.
Conclusions
Lines 389-392. “despite having presented a higher elevation of SBP and DBP in relation to PL, 24 hours later, it promised a hypotensive effect, with lower absolute values for the CA condition in SBP, and DBP”. According to the results, there is no significant increase in either SBP or DBP in AC training in relation to PL. Therefore, this cannot be affirmed. 24 hours later there are no significant differences in DBP between CA and PL. Therefore, it cannot be stated that there is a hypotensive effect.
Line 393, “However, it showed a reduction after 40min”.
The interpretation of the results I make is that at min 40 there are significant differences and also at minute 60. And at minute 50 is where the HR value is lowest when supplemented with PL, (I don't understand why there is no significant difference). So, HR was reduced with CA, but even more with PL.
It is necessary to modify the conclusions.
Reviewer 3 Report
I put comments throughout the documents. My comments are grammatical only.
Do not begin a sentence with a number, spell out the numbers and all numbers in the same sentence should be spelled out.
Use scientific abbreviations for time and be consistent, Min should be min always, not sometimes MIN and sometimes min
I have never seen min abbreviated '
Any abbreviation used in Fig and Tables should be explained in the legend, like '
A couple of sentences were unclear, reread and rewrite sentences
1. Introduction
We have the following hypoth.... a) would caffeine increase .... should say it was hypothesized that caffeine woul..... b)... etc.
4. Discussion
The present study aimed to...rewrite ......the present study analyzed
Your questions should be hypotheses, our hypotheses were.... and should not be stated as questions
I do not understand what you mean by heating in Fig 1
All units should be stated in the table 1RM (kg)/body mass (kg)
Were all side effects measured on the Quest scale-state this explicitly
What is meant by Increased activity and force?
In figure 2, Min should always be min, use scientific abb for Hours

Round 2
Reviewer 2 Report
The reviewers' responses have not addressed my concerns. The results are unclear and complicated for the reader. Furthermore, the discussion and conclusions drawn are not in line with the results.